# Potassium Ions Decrease Mitochondrial Matrix pH: Implications for ATP Production and Reactive Oxygen Species Generation

**DOI:** 10.3390/ijms25021233

**Published:** 2024-01-19

**Authors:** Jannatul Naima, Yoshihiro Ohta

**Affiliations:** Department of Biotechnology and Life Science, Tokyo University of Agriculture and Technology, Nakacho, Koganei, Tokyo 184-8588, Japan; s207804z@st.go.tuat.ac.jp

**Keywords:** mitochondria, potassium ions, ATP production, matrix pH, ROS generation

## Abstract

Potassium (K^+^) is the most abundant cation in the cytosol and is maintained at high concentrations within the mitochondrial matrix through potassium channels. However, many effects of K^+^ at such high concentrations on mitochondria and the underlying mechanisms remain unclear. This study aims to elucidate these effects and mechanisms by employing fluorescence imaging techniques to distinguish and precisely measure signals inside and outside the mitochondria. We stained the mitochondrial matrix with fluorescent dyes sensitive to K^+^, pH, reactive oxygen species (ROS), and membrane potential in plasma membrane-permeabilized C6 cells and isolated mitochondria from C6 cells. Fluorescence microscopy facilitated the accurate measurement of fluorescence intensity inside and outside the matrix. Increasing extramitochondrial K^+^ concentration from 2 mM to 127 mM led to a reduction in matrix pH and a decrease in the generation of highly reactive ROS. In addition, elevated K^+^ levels electrically polarized the inner membrane of the mitochondria and promoted efficient ATP synthesis via FoF_1_-ATPase. Introducing protons (H^+^) into the matrix through phosphate addition led to further mitochondrial polarization, and this effect was more pronounced in the presence of K^+^. K^+^ at high concentrations, reaching sub-hundred millimolar levels, increased H^+^ concentration within the matrix, suppressing ROS generation and boosting ATP synthesis. Although this study does not elucidate the role of specific types of potassium channels in mitochondria, it does suggest that mitochondrial K^+^ plays a beneficial role in maintaining cellular health.

## 1. Introduction

Mitochondria are intracellular organelles responsible for producing ATP, cellular energy, through an oxidative phosphorylation process [1,2,3]. Mitochondria are also the primary production sites of reactive oxygen species (ROS) as byproducts of oxidative phosphorylation. ROS function as signaling molecules at low concentrations; however, they can be detrimental to cells at high concentrations by oxidizing cellular components, such as proteins, lipids, and nucleic acids [4,5,6]. Therefore, it is essential for cells to efficiently produce ATP while suppressing the generation of ROS to low levels to ensure cell survival.

ATP and ROS production in mitochondria are intrinsically linked to their membrane potential (ΔΨ_m_) and matrix pH. Consequently, these parameters have been extensively measured in numerous studies exploring the mechanisms of ATP synthesis and ROS generation [7,8,9,10]. Fluorescence imaging has emerged as an effective technique for examining these factors with high accuracy and has been successfully employed [11,12]. In this method, the mitochondrial matrix is stained with pH-sensitive dyes or dyes responsive to membrane potential changes, allowing for visualization of their distribution. This approach facilitates the differentiation of signals between intramitochondrial and extramitochondrial environments, enabling precise measurements. On the other hand, fluorescence measurements of cellular or mitochondrial suspensions are less accurate, as they capture mixed signals from inside and outside the mitochondria. Electrodes have been another method employed to assess membrane potential and pH [13]. Here, membrane potential is gauged by measuring the concentration of permeable, positively charged molecules in the solution, whereas pH is determined by quantifying H^+^ concentration. For this reason, electrode-based methods predominantly measure concentrations outside the mitochondria, providing only indirect insights into internal changes. This limitation is particularly pronounced in pH measurements, as the internal buffering capacity of the mitochondria complicates the determination of intramitochondrial pH levels.

Potassium (K^+^) is the most abundant cation in the cytosol, existing at high concentrations of around 140 mM. This concentration is approximately ten times that of sodium (Na^+^) and more than a million times higher than that of calcium (Ca^2+^) and protons (H^+^) [14]. It is thought that K^+^ is also present in high concentrations within the mitochondrial matrix [15,16]. In mitochondria, several types of K^+^ channels exist [17,18], through which K^+^ enters the matrix and is expelled by the K^+^/H^+^ exchanger (KHE) [19,20]. Mitochondria maintain K^+^ concentration within the matrix by regulating the activities of these channels and exchangers. The effects of K^+^ on mitochondria have been studied using K^+^ ionophores and compounds that act on K^+^ channels (inhibitors and openers) [21,22,23]. However, in the presence of ionophores, an excessive influx of K^+^ disrupts the mitochondrial membrane potential (ΔΨ_m_) [24]. This disruption of ΔΨ_m_ affects various mitochondrial functions, meaning that changes observed in mitochondrial functions may be attributable to changes in ΔΨ_m_ rather than to alterations in K^+^ concentration within the matrix. Furthermore, determining the relationship between changes in mitochondrial activity and variations in matrix K^+^ concentration is challenging when using specific inhibitors or openers of K^+^ channels. This difficulty arises for two main reasons. Firstly, since various K^+^ channels exist in mitochondria, how these compounds affect the K^+^ concentration in the matrix is unclear. Secondly, changes in mitochondrial activity observed with these inhibitors or openers may not be directly related to shifts in K^+^ concentration. Instead, they could be associated with alterations in proteins functionally linked to the targeted channels, as suggested in previous research [25]. Thus, understanding the effects of matrix K^+^ concentration on mitochondrial function remains a complex task. Indeed, the effects of matrix K^+^ on ATP synthesis and ROS generation remain debated. Some studies suggest that K^+^ enhances ATP production when transported into the matrix via the FoF_1_-ATPase [26], whereas others report decreased ATP production [27] or no significant effect on ATP synthesis [28]. Also, many studies have investigated the role of K^+^ transport in mitochondrial ROS production [23,29,30]. As with the effects on ATP synthesis, the results are controversial, with some finding an increase in ROS with activation of K^+^ channels [22,31] and others finding a decrease in ROS [21,32].

In this study, to elucidate the effects of K^+^ on mitochondrial function and the mechanisms involved, we examine how K^+^ at sub-hundred millimolar or higher concentrations in the matrix, maintained via K^+^ channels, affects ATP synthesis and ROS generation. A notable aspect of our methodology is the introduction of K^+^ into the matrix under mild conditions without using K^+^ channel openers or K^+^ ionophores to avoid the effects of rapid influx. Subsequently, we precisely measured the mitochondrial response to respiratory substrates by distinguishing and evaluating signals originating from inside and outside the matrix. Our findings indicate that K^+^ at these elevated levels in the matrix promotes ATP synthesis and inhibits ROS generation by dissociating H^+^ from weak acids. These results suggest a novel role for K^+^, maintained at high concentrations in the matrix through potassium channels, in supporting mitochondrial functions.

## 2. Results

### 2.1. Effects of K^+^ on ROS Generation

To examine the effects of K^+^ on ROS generation in the mitochondria, the fluorescence of MitoSOX Red in plasma membrane-permeabilized C6 cells was observed (Figure 1A; Appendix A). The fluorescence images were analyzed by calculating the integrated fluorescence intensity of MitoSOX Red in individual cells (F_MSR_).

When cells were incubated in a Tris-0K buffer without ADP and Pi, adding 70 mM K^+^ drastically suppressed mitochondrial ROS generation (Figure 1B; Appendix A). A further increase in the K^+^ concentration to 125 mM did not affect this. In the buffer containing ADP and Pi, ROS generation was significantly low and unaffected by K^+^. Adding oligomycin to the buffer with ADP and Pi significantly increased ROS generation in the mitochondria. However, this increase is counteracted by the presence of K^+^. Our results reveal that (1) K^+^ significantly decreased ROS generation in mitochondria when F_O_F_1_-ATPase was inhibited, and (2) the presence of ADP and Pi affected ROS generation in mitochondria in F_O_F_1_-ATPase-dependent and independent manners.

### 2.2. Effects of K^+^ on Matrix pH

Since the H^+^ concentration in the matrix is a crucial factor for regulating ROS generation in mitochondria, we examined the effects of K^+^ on matrix pH. To measure matrix pH, permeabilized cells of the plasma membrane were stained with the pH-sensitive fluorescent dye BCECF and observed using fluorescence microscopy (Figure 2A; Appendix A). The fluorescence pattern of BCECF was consistent with that of TMRE, indicating that the BCECF signal was exclusively from mitochondria.

When cells were in a buffer without ADP and Pi or with ADP, Pi, and oligomycin, an increase in K^+^ concentration from 0 to 70 mM or from 2 to 72 mM substantially decreased the matrix pH (Figure 2B); a further increase in K^+^ concentration did not reduce the matrix pH. In contrast, K^+^ did not lower the matrix pH when K^+^ was added to the cells in the buffer containing ADP and Pi without oligomycin (Figure 2B). The response of matrix pH to K^+^ was similar to that of ROS generation. This is consistent with a previous finding that matrix pH is a crucial determinant of ROS generation in mitochondria [24].

### 2.3. Characterization of K^+^ Influx into Mitochondria

Next, we examined whether K^+^ entered the mitochondrial matrix. Plasma membrane-permeabilized cells were stained with the K^+^-sensitive fluorescent dye PBFI and observed using fluorescence microscopy. The detailed pattern of PBFI fluorescence completely coincided with that of the TMRE fluorescence, indicating that the signal was from mitochondria (Figure 3A; Appendix A). The fluorescence images were analyzed by calculating the integrated fluorescence intensity of PBFI in individual cells (F_PBFI_). When K^+^ concentrations were increased from 10 to 70 mM, F_PBFI_ considerably increased (Figure 3B,C). The increase in F_PBFI_ upon K^+^ addition was diminished by CCCP addition (Figure 3D,E). However, even considering PBFI was 1.5 times more selective for K^+^ than Na^+^ (Thermo Fisher Scientific), Na^+^ did not increase F_PBFI_ at 70 mM (Figure 3B,C). These results are consistent with the previous findings that K^+^ selectively entered the mitochondrial matrix in an electrical gradient (ΔΨ_m_)-dependent manner [23].

Next, we investigated the contribution of mitoK_ATP_ (mitochondrial ATP-sensitive potassium channels), the most studied mitochondrial potassium channels [17,23,33], to K^+^ entry. To this end, glibenclamide, an inhibitor of mitoK_ATP_ channels, was added to mitochondria. Glibenclamide did not completely inhibit K^+^ entry (Appendix A) but strongly inhibited it in the presence of CCCP (Appendix A), which confirmed that glibenclamide inhibited the potassium entry in an electron gradient-independent manner. These results suggest that several K^+^ channels, including the mitoK_ATP_ channel, contributed to the observed K^+^ entry into the matrix.

### 2.4. Effects of K^+^ on ATP Production of Mitochondria

To investigate the effects of K^+^ on mitochondrial ATP production, we measured ATP production in buffers containing either 2 mM K^+^, 72 mM K^+^, or a combination of 2 mM K^+^ and 70 mM Na^+^. These results are depicted in Figure 4. The ATP production in the buffer with 72 mM K^+^ was significantly higher than in the buffers with either 2 mM K^+^ alone or 2 mM K^+^ combined with 70 mM Na^+^ (Figure 4A). Conversely, no significant differences in ATP production were observed among the three buffers when oligomycin was present. The ATP production attributable to F_O_F_1_-ATPase was calculated by subtracting the ATP level produced in the presence of oligomycin from that produced in its absence, as shown in Figure 4B. These findings indicate that increasing the K^+^ concentration from 2 to 72 mM enhanced ATP production via oxidative phosphorylation.

### 2.5. Effects of K^+^ on ΔΨ_m_

K^+^ decreased the pH gradient across the inner membrane but stimulated mitochondrial ATP production. Considering that ATP production relies on a proton motive force (PMF) composed of the mitochondrial membrane potential (ΔΨ_m_) and the pH gradient, we investigated the effect of K^+^ on ΔΨ_m_, another component of the PMF. For this, we analyzed TMRE fluorescence in plasma membrane-permeabilized cells (Figure 5A,B). To compare TMRE fluorescence intensity under various conditions, we integrated TMRE fluorescence over individual cells [34]. This integrated TMRE fluorescence intensity in individual cells is referred to as F_TMRE_.

Figure 5C shows the changes in F_TMRE_ upon adding MG and oligomycin. In all conditions, regardless of the presence of Na^+^ and K^+^, F_TMRE_ increased upon adding MG. Furthermore, the subsequent addition of oligomycin, a specific F_O_F_1_-ATPase inhibitor, led to a further increase in F_TMRE_ (Figure 5C,D). These findings suggest that MG addition caused mitochondrial polarization by stimulating proton pumping through the electron transfer chain. The subsequent oligomycin addition further enhanced polarization by inhibiting proton entry into the matrix via F_O_F_1_-ATPase.

Figure 5D shows F_TMRE_ before and after the addition of MG and the further addition of oligomycin. As shown in Figure 5D, before the addition of oligomycin, no significant difference in F_TMRE_ was observed in a buffer containing 2 mM K^+^ or a buffer containing 72 mM K^+^. In contrast, in the presence of oligomycin, F_TMRE_ in a buffer with 72 mM K^+^ was higher than in a buffer with 2 mM K^+^. This result indicates that the mitochondria in the buffer with 72 mM K^+^ were more polarized than those in the buffer with 2 mM K^+^. To verify this, we analyzed the increase in F_TMRE._ The rise in F_TMRE_ (ΔF_TMRE_) upon adding oligomycin was significantly greater in the buffer with 72 mM K^+^ compared to the buffer with 2 mM K^+^ (Figure 5E). This indicates that proton entry into the matrix through F_O_F_1_-ATPase was enhanced in the presence of 72 mM K^+^. These findings suggest that 72 mM K^+^ stimulated proton pumping from the matrix to the cristae space and enhanced ATP production by F_O_F_1_-ATPase. Although the K^+^ concentration tested (72 mM) was lower than the physiological concentration (140 mM), these results indicate that 72 mM K^+^ sufficiently stimulated proton pumping. Unlike K^+^, adding Na^+^ in the buffer decreased the changes in F_TMRE_. Consistent with the ATP production results, 70 mM Na^+^ significantly reduced mitochondrial polarization.

Similar results were obtained for isolated mitochondria (Appendix A), supporting the idea that the TMRE fluorescence changes observed for plasma membrane-permeabilized cells were due to mitochondrial characteristics.

### 2.6. Effects of the Increase in Matrix H^+^ Concentration on ΔΨ_m_

In the presence of oligomycin, elevating the K^+^ concentration in the buffer from 2 to 72 mM significantly increased H^+^ concentration within the mitochondrial matrix and induced mitochondrial polarization. To investigate the relationship between H^+^ concentration in the matrix and mitochondrial polarization, we introduced H^+^ into the matrix by adding phosphate to the buffer and measured F_TMRE_. The phosphate entered the matrix in an electrically neutral state, accompanied by H^+^, thereby dissociating H^+^ within the matrix [35].

These results are shown in Figure 6A,B. Adding MG resulted in the polarization of mitochondria in both the presence and absence of K^+^. The degree of polarization was unaffected by K^+^ concentration. However, the subsequent addition of phosphate (1 mM KH_2_PO_4_) further polarized the mitochondria in both buffer types. This polarization effect was more pronounced in mitochondria with 70 mM K^+^. These observations align with the hypothesis that increased H^+^ concentration within the matrix leads to mitochondrial polarization.

### 2.7. Effects of K^+^ on the Buffer pH

The increase in K^+^ concentration led to a rise in the matrix H^+^ concentration and induced mitochondrial polarization. To investigate the mechanism behind the K^+^-induced increase in matrix H^+^ concentration, we added K^+^ to various solutions and measured pH changes. Phosphate buffer and bovine serum albumin (BSA) solutions were chosen as the buffer and condensed protein solutions, respectively. In both cases, adding 70 mM K^+^ (for a total of 72 mM for phosphate buffer) significantly lowered the pH (Figure 7A,B). Similarly, the pH markedly decreased when K^+^ was added to Tris or HEPES buffer at 70 mM regardless of BSA presence (Appendix A). These results are consistent with observations that K^+^ entry into the matrix reduced matrix pH. The pH decreases can be attributed to the increased dissociation constant of weak acids at high ionic strength [36,37]. Although adding K^+^ lowered the buffer pH, this minor decrease in pH outside the mitochondria did not impact ATP synthesis within the mitochondria (Appendix A).

## 3. Discussion

In this study, increasing the K^+^ concentration in the buffer surrounding the mitochondria to sub-hundred millimolar levels decreased matrix pH and ROS production, induced mitochondrial polarization, and stimulated ATP production.

The observed pH change could be explained by two potential mechanisms. The first mechanism suggests that the entry of K^+^ into the matrix elevates its ionic strength, which, in turn, facilitates the dissociation of H^+^ from weak acids [36,37], thereby increasing the H^+^ concentration in the matrix. The second mechanism involves K^+^/H^+^ exchange mediated by KHE. The expulsion of K^+^ from the matrix is accompanied by an influx of H^+^ into the matrix, resulting in a reduction in matrix pH.

Matrix pH is a crucial factor regulating ROS generation in mitochondria. A decrease in matrix pH (an increase in H^+^ concentration in the matrix) should enhance electron transfer, coupled with the proton translocation from the matrix to the cristae space. Since ROS generation is known to increase when electron transfer is inhibited [38], a decrease in matrix pH suppresses ROS generation. Moreover, as ubisemiquinone, a free radical, converts to ubiquinol in the presence of H^+^, the lowered matrix pH reduces free radicals and suppresses ROS production. Previous studies [24,39] reported that K^+^ entry into the matrix increased the matrix pH. These studies utilized valinomycin, a K^+^ ionophore, to stimulate K^+^ entry into the matrix. Under these conditions, ΔΨ_m_ collapsed due to excessive K^+^ entry, leading to an over-ejection of protons from the matrix and an increase in matrix pH. However, in the present experiments, unlike previous ones, K^+^ entry into the matrix occurred solely through the mitochondrial K^+^ channel, preventing excessive K^+^ influx. The effect of K^+^ on matrix pH was observed in mitochondria, where there was a balance between K^+^ efflux and influx.

The increase in H^+^ concentration in the matrix also stimulated mitochondrial polarization. As the increase in matrix H^+^ concentration led to a diminished pH gradient across the inner membrane (ΔpH), this could have prompted the proton pumps to eject more protons in compensation, resulting in further polarization of the inner membrane. We discussed two mechanisms by which K^+^ decreases the pH in the mitochondrial matrix. However, among these mechanisms, we believe that promoting proton dissociation from weak acids by K^+^ is at least necessary for the K^+^-induced polarization of mitochondria. This is because the impact of K^+^ on mitochondrial polarization was not observed when phosphate levels were low but was limited to conditions with high phosphate concentrations. This implies that in the presence of low concentrations of weak acids, such as phosphate, within the matrix, K^+^ cannot supply sufficient protons to the matrix for effective mitochondrial polarization.

We observed that K^+^ enhanced mitochondrial polarization and ATP production. There are two potential ways that the observed mitochondrial polarization could have augmented ATP production. The first is an increase in PMF. Assuming that the proton ejection from the matrix depends on the pH of the matrix in addition to the difference in membrane potential (ΔΨ_m_) and the pH (ΔpH) across the inner membrane, the PMF will increase when the pH of the matrix is lower. The second mechanism is the ADP/ATP exchange stimulation by the membrane potential-driven adenine nucleotide translocator.

Based on the above considerations, we propose the model depicted in Figure 8. This model comprises three principal components: (1) K^+^ entry into the matrix through K^+^ channels reduces the matrix pH, (2) the increased concentration of H^+^ suppresses the generation of ROS, and (3) this increase in H^+^ concentration in the matrix also polarizes mitochondria, subsequently increasing ATP production. In conditions where K^+^ is absent, the proton-pumping activities of the electron transfer chains (ETCs) and FoF_1_-ATPase (ATP synthase) are diminished due to a reduced H^+^ concentration in the mitochondrial matrix (Figure 8A). This decrease in proton availability leads to a suppression of ATP synthesis and an increase in the production of ROS. On the other hand, when the concentration of K^+^ in the matrix reaches sub-hundred millimolar or higher, K^+^ augments the H^+^ concentration within the mitochondrial matrix (Figure 8B). This increase occurs through two primary mechanisms: the entry of H^+^ via KHE and the dissociation of protons from weak acids. Subsequently, this elevated H^+^ concentration stimulates proton pumping by ETCs and accelerates the conversion of ubisemiquinone to ubiquinol. These processes collectively contribute to an increase in ATP production and a reduction in the generation of ROS.

Halestrap et al. found that matrix swelling, induced by K^+^, enhances substrate oxidation [40]. However, the precise mechanism through which this swelling facilitates substrate oxidation is not fully understood. Our results may explain their findings. We observed that K^+^ at sub-hundred millimolar concentrations or higher stimulated H^+^ translocation from the matrix to the cristae space, necessitating increased substrate oxidation. Therefore, it is plausible that K^+^’s role in promoting H^+^ translocation is linked to the enhanced substrate oxidation observed by Halestrap et al. [40].

In the present study, we observed significant effects of K^+^ at a concentration of 70 mM, which was below the physiological concentration. The matrix pH remained stable when the K^+^ concentration increased from 70 to 125 mM. Based on these observations, we hypothesize that the effects we identified are likely to be relevant at physiological K^+^ concentrations as well. However, under the conditions used in this study, the mitochondrial membrane potential became unstable when the K^+^ concentration in the buffer was increased to physiological levels while maintaining physiological osmolality. Therefore, the effect of physiological intracellular K^+^ concentration on membrane potential and ATP production was not strictly observed. In addition, due to the small dissociation constant of the K^+^-sensitive fluorescent dye, it was not possible to observe the behavior of K^+^ in the mitochondria when the K^+^ concentration in the buffer was increased from sub-hundred millimolar to physiological levels. Research to overcome these problems is positioned as a future topic.

## 4. Materials and Methods

### 4.1. Materials

The C6 rat glioma cell line (accession number: CVCL 0194) was purchased from Riken Cell Bank (Wako, Japan). Roswell Park Memorial Institute Medium 1640 (RPMI 1640; catalog number (Cat#): 31800022) was obtained from Gibco (Grand Island, NY, USA). Tetramethylrhodamine ethyl ester (TMRE) (Cat#: T669), MitoSox Red (Cat#: M36008), and 2′,7′-Bis(carboxyethyl)-5,6-carboxyfluorescein (BCECF) acetoxymethyl ester (BCECF-AM) (Cat#: B1170) were purchased from Thermo Fisher Scientific (Waltham, MA, USA). PBFI-AM (Cat#: P-1267) was purchased from Invitrogen (Carlsbad, CA, USA). CellTiter Glo (Cat#: G7571/2/3), BCA protein assay kit (Cat#: 23225), and digitonin (Cat#: 043-21376) were purchased from Promega (Fitchburg, WI, USA), Thermo Scientific (Rockford, IL, USA), and Sigma-Aldrich Co. (St. Louis, MO, USA), respectively. All other chemicals used were of the highest available purity.

### 4.2. Cell Culture, Plasma Membrane Permeabilization, and Mitochondria Isolation

C6 rat glioma cells were cultured in RPMI 1640 medium, supplemented with 10% fetal bovine serum, at 37 °C in a humidified atmosphere containing 5% CO_2_. The cells are noted for their ease of proliferation and are convenient for studies involving various aspects of mitochondrial activity. For microscopic examination, C6 cells were grown on collagen-coated, glass-bottomed culture dishes for 2–3 days before use. 

To control the environment surrounding the mitochondria and to monitor their direct response, the plasma membranes of C6 cells were permeabilized. For this purpose, C6 cells were incubated with 30 µM digitonin in Tris-0K buffer (composed of 10 mM Tris-HCl, 250 mM sucrose, 0.5 mM EGTA, at pH 7.4) for 3 min at 4 °C [41]. Following this, the cells were washed three times with the same Tris-0K buffer. Permeabilization of the plasma membranes was achieved through gentle pipetting [42]. In some experiments, isolated mitochondria were utilized in place of intracellular mitochondria. The procedure for isolating mitochondria from C6 cells involved initially permeabilizing the plasma membranes with digitonin as described, followed by washing. Subsequently, the cells were detached from the dish by pipetting, and the mitochondria released from cells in this process were collected through differential centrifugation [42] and suspended in Tris-0K buffer. All steps for isolating mitochondria were carried out at 4 °C.

The protein concentration of the mitochondrial suspension was determined using a BCA protein assay kit. For the microscopic observation of individual isolated mitochondria, a 1 mL mitochondrial suspension (0.03 mg protein/mL) was centrifuged on a glass-bottomed culture dish (35 mm in diameter) at 100× *g* for 5 min at 4 °C. This was performed to adsorb mitochondria onto the dish. Subsequently, the adsorbed mitochondria were washed thrice with the Tris-0K buffer.

### 4.3. Detection of ROS Production

To monitor reactive oxygen species (ROS) production in mitochondria, we stained mitochondria in C6 cells with MitoSOX Red, a mitochondrial-specific fluorescence indicator of superoxide anion [43]. For staining, plasma membrane-permeabilized C6 cells were incubated with 2.5 µM MitoSOX Red for 10 min at 25 °C in various buffers. These buffers included Tris-0K buffer, Tris-70K buffer (10 mM Tris-HCl, 70 mM KCl, 110 mM sucrose, 0.5 mM EGTA, pH 7.4), and Tris-125K buffer (10 mM Tris-HCl, 125 mM KCl, 0.5 mM EGTA, pH 7.4). For the measurements of MitoSOX fluorescence in the presence of phosphate and ADP, 1 mM KH_2_PO_4_ and 0.5 mM KH_2_ADP were supplemented in these buffers. In oligomycin experiments, it was added at a concentration of 1 μM. The final K^+^ concentrations in the buffers, all of which contained 1 mM KH_2_PO_4_ and 0.5 mM KH_2_ADP, were 2 mM in the Tris-0K buffer, 72 mM in the Tris-70K buffer, and 127 mM in the Tris-125K buffer. Sucrose was present in the Tris-70K buffer to ensure consistent osmolality across all the buffers.

After the 10 min incubation with MitoSOX Red, the samples were transferred to the stage of an inverted epifluorescence microscope (IX-73; Olympus; Tokyo, Japan). The samples were visualized using a 10× objective lens (UPlanXApo, NA = 0.4; Olympus) with an excitation light with a wavelength range of 510–550 nm produced by a 75 W xenon lamp. Emissions > 580 nm were captured using a cooled CCD camera (MD-695, Molecular Device Japan; Tokyo, Japan) with 2 × 2 binning pixels. Each frame was exposed for 1 s. To avoid photodynamic damage to the mitochondria, the illumination intensity was reduced to 1.5% using a neutral-density filter. Finally, the fluorescence signals from mitochondria in permeabilized cells were recorded at 25 °C, digitized to 14 bits, and analyzed using image-processing software (MetaMorph Ver.7.8; Universal Imaging; Downingtown, PA, USA).

### 4.4. Measurements of pH in the Mitochondrial Matrix

To examine pH changes in the mitochondrial matrix, we entrapped BCECF, a fluorescent pH indicator [44], in the matrix. First, the plasma membrane-permeabilized C6 cells were incubated with 5 µM BCECF-AM for 30 min at 25 °C in the buffer as described in Section 4.3. The cells were then washed three times with the buffer. BCECF was excited with a light wavelength between 390 and 420 nm (shorter wavelength) or between 470 and 490 nm (longer wavelength). BCECF fluorescence was recorded between 515 and 550 nm. Fluorescence images of the dye illuminated with shorter wavelengths (F_S_) and longer wavelengths (F_L_) were obtained. When intracellular distributions of BCECF were observed, BCECF fluorescence was acquired with a 40× objective lens (Uapo40×/340, NA = 0.9; Olympus). The other experimental settings used for the BCECF fluorescence observations were identical to those used for the MitoSOX Red.

To determine the matrix pH based on the fluorescence ratio (F_L_/F_S_), BCECF fluorescence ratios were measured at multiple matrix pH values, and a calibration curve was generated. The pH of the matrix was regulated by adjusting the pH of the buffer containing 5 µM carbonyl cyanide m―chlorophenyl hydrazine (CCCP). Calibration curves for pH measurement were obtained using the same buffer as employed for determining the unknown pH in the matrix. The data were analyzed by fitting the equation (F_L_/F_S_) = (A + B × 10^(7-pH)^)/(C + 10^(7-pH)^) using the least-squares method, as previously reported by James-Kracke [45] (Appendix A).

### 4.5. Measurement of K^+^ Influx into Mitochondria

We used PBFI, a K^+^-sensitive fluorescence indicator, to visualize the K^+^ influx into the mitochondrial matrix. To achieve this, we permeabilized the plasma membranes of C6 cells and incubated them with 10 µM PBFI-AM in Tris-0K buffer with 10 mM KCl at 25 °C for 30 min. After incubation, the cells were washed thrice with the same buffer. PBFI was excited using a 340–380 nm wavelength. Emissions greater than 500 nm were captured in a series of image frames acquired at intervals of 1 min with 2 × 2 binning pixels. The acquisition was computer controlled to estimate the time-resolved fluorescence. Each frame was exposed for 1 s, and the excitation light was cut off using a mechanical shutter for the remaining 59 s to prevent mitochondrial damage from prolonged illumination. The other experimental settings used for the PBFI fluorescence observation were identical to those used for the BCECF fluorescence observation, as described in Section 4.4 [33].

### 4.6. Measurement of ATP in Isolated Mitochondria

Before assessing ATP production in mitochondria, isolated mitochondria were resuspended in either Tris-0K, Tris-70K, or Tris-NaCl buffer (10 mM Tris-HCl, 70 mM NaCl, 110 mM sucrose, 0.5 mM EGTA, pH 7.4) to a concentration of 0.06 mg protein/mL suspension. ATP synthesis was initiated by adding 0.5 mM KH_2_ADP, 1 mM KH_2_PO_4_, 5 mM malate, and 5 mM glutamate to the mitochondrial suspension. This suspension was then incubated at 25 °C for 10 min, followed by centrifugation at 8000× *g* for 10 min at 4 °C. The supernatant was collected, and the ATP concentration was measured using the CellTiter-Glo^®^ Luminescent Cell Viability Assay kit following the manufacturer’s instructions (Promega, Fitchburg, WI, USA).

### 4.7. Fluorescence Imaging of Changes in Mitochondrial Membrane Potential (ΔΨ_m_)

To observe changes in ΔΨ_m_, TMRE, a fluorescent dye accumulating in mitochondria in a membrane potential-dependent manner, was used [24]. Plasma membrane-permeabilized C6 cells and mitochondria isolated from C6 cells were stained with 20 nM and 10 nM TMRE, respectively. Staining was performed by incubation at 25 °C for 10 min in a buffer containing 1 mM KH_2_PO_4_, 0.5 mM KH_2_ADP, and 1 mg/mL BSA [34]. The buffers used were Tris-0K buffer, Tris-70K buffer, or Tris-NaCl buffer, as described in Section 4.6, because ΔΨ_m_ was not stable for more than 10 min in the buffers without sucrose. The samples were illuminated with excitation light with a 510–550 nm wavelength range. Emissions > 580 nm were captured. The other experimental settings used for the TMRE fluorescence observation were identical to those used for the PBFI fluorescence observation, as described in Section 4.5 [34].

### 4.8. Analysis of Fluorescence Images

To analyze the fluorescence intensity of the cells, we integrated the fluorescence intensity of MitoSOX Red, BCECF, PBFI, and TMRE over the entire cell. The integrated fluorescence intensity was then calculated using a method previously described [34]. Cells not in contact with adjacent cells were selected to measure the whole-cell fluorescence, and the integrated fluorescence intensity was calculated. For each isolated mitochondrion, the average fluorescence intensity was calculated across a 0.6 μm^2^ area on the mitochondrion. The fluorescence intensity of TMRE in the buffer was measured in the same field as the mitochondria at a location where mitochondrial TMRE did not affect the measurement [24]. The TMRE fluorescence intensity in the buffer was subsequently subtracted from the TMRE fluorescence intensity of the mitochondria.

### 4.9. Statistical Analysis

We averaged the results of at least three independent experiments employing isolated mitochondria and permeabilized C6 cells. The results were expressed as mean ± standard error of the mean (SEM). Data were analyzed using a two-tailed analysis of variance, followed by the Student–Newman–Keuls test. Values of *p* < 0.05 were considered to indicate statistically significant differences.

## 5. Conclusions

The effects of K^+^ on mitochondria have been extensively studied for a long time. Although the concentration of K^+^ in the cytosol is the highest of all cations, the specific impact of K^+^ on mitochondria at such high cytosolic concentrations and the mechanisms behind this remain unclear. This study addresses these questions. We found that when the concentration of K^+^ around the mitochondria was at sub-hundred millimolar levels or higher, K^+^ promoted the dissociation of protons from weak acids, decreasing matrix pH. As a result, ATP synthesis was enhanced and the generation of ROS was suppressed. Such effects of K^+^ suggest that high cytosolic K^+^ concentrations are crucial to sustaining cellular health. Although our study did not directly explore the effects of physiological fluctuations in K^+^ concentration on mitochondrial functionality, it highlights the importance of maintaining K^+^ concentrations at or above sub-hundred millimolar levels within cells.

## Figures and Tables

**Figure 1 ijms-25-01233-f001:**
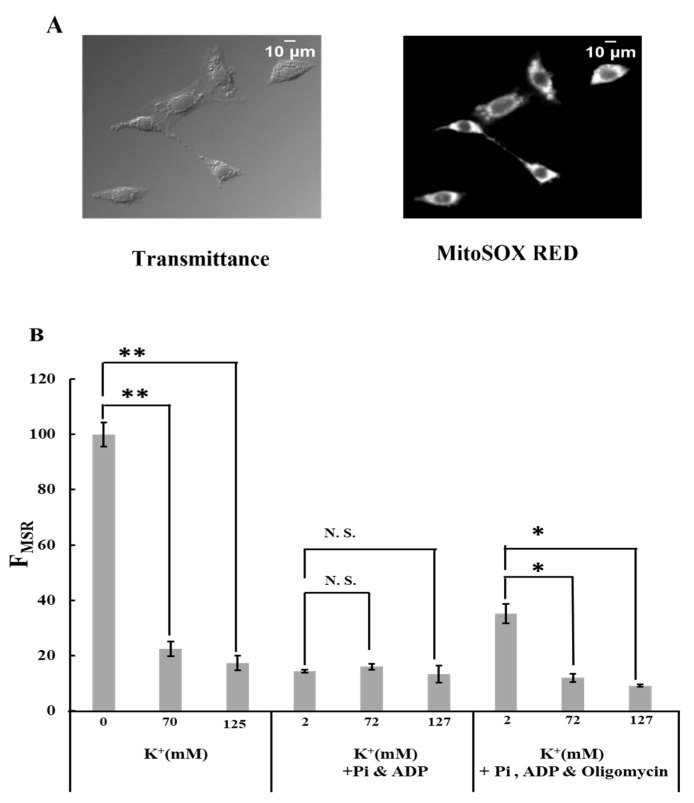
Effects of K^+^ on ROS generation: (**A**) transmittance and MitoSOX Red fluorescence images of permeabilized C6 cell. The transmittance image was obtained using differential interference optical microscopy. Bar, 10 μm. (**B**) MitoSOX Red fluorescence intensity. The intensity was measured in Tris-0K buffer, Tris-70K buffer, and Tris-125K buffer after adding 5 mM malate and 5 mM glutamate. In some experiments, MitoSOX Red fluorescence was measured in the buffer with Pi (1 mM KH_2_PO_4_) and ADP (0.5 mM KH_2_ADP) or the buffer with Pi, ADP, and oligomycin (1 μΜ). The fluorescence value in Tris-0K buffer without Pi, ADP, and oligomycin was normalized to 100. Three independent experiments were performed for each condition, with 15 cells analyzed in each experiment. Values represent the mean ± SEM (n > 3). N.S., not significant; ** *p* < 0.01; * *p* < 0.05 vs. buffer with 0 or 2 mM K^+^ for the respective conditions.

**Figure 2 ijms-25-01233-f002:**
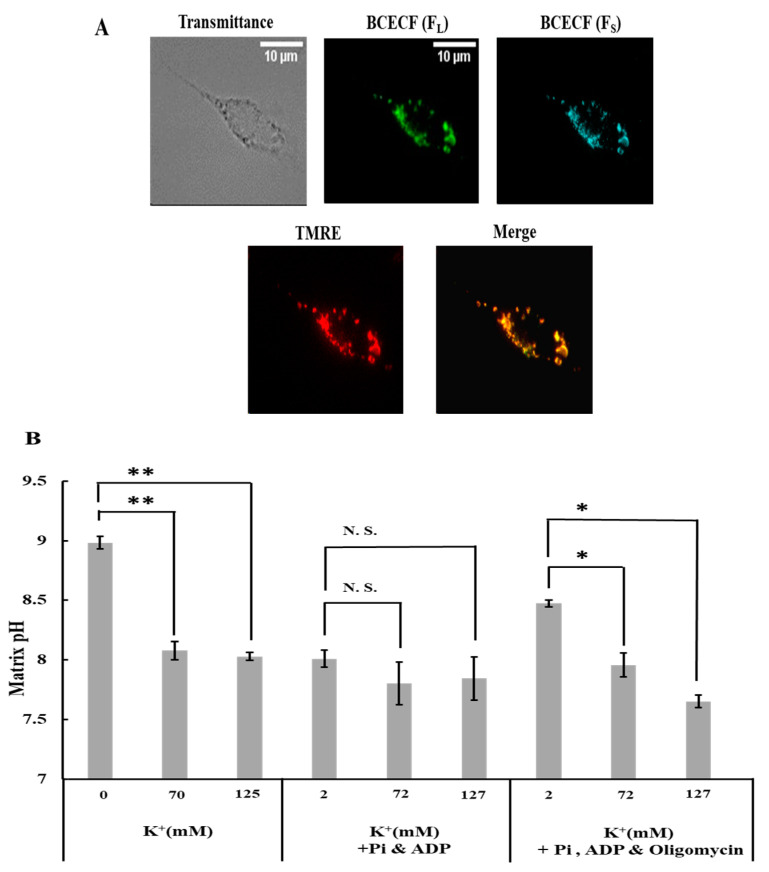
Effects of K^+^ on mitochondrial matrix pH: (**A**) plasma membrane permeabilized C6 cells stained with BCECF and TMRE. Bar, 10 μm. (**B**) Matrix pH under various conditions. The conditions used are the same as those explained in Figure 1B. Values represent the mean ± SEM (n > 3). N.S., Not significant; ** *p* < 0.01; * *p* < 0.05 vs. buffer with 0 or 2 mM K^+^ for the respective conditions.

**Figure 3 ijms-25-01233-f003:**
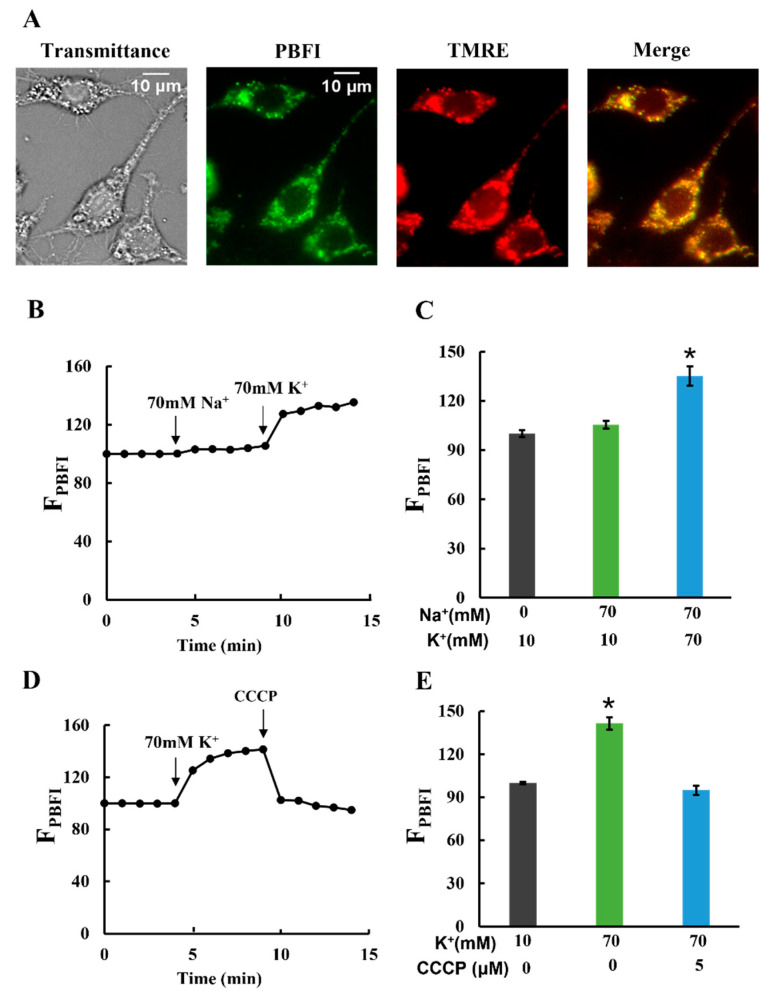
K^+^ influx into mitochondria: (**A**) double staining of mitochondria in permeabilized C6 cells with PBFI and TMRE. The transmittance image was obtained with phase contrast microscopy. Bar, 10 μm. (**B**–**E**) Changes in PBFI fluorescence in the matrix (F_PBFI_). The initial value was normalized to 100. (**B**,**C**) A total of 70 mM Na^+^ and 70 mM K^+^ were added at t = 4 and 9 min, respectively. (**D**,**E**) A total of 70 mM K^+^ and 5 μM CCCP were added at t = 4 and 9 min, respectively. Three independent experiments were performed for each condition, with 15 cells analyzed in each experiment. Values represent the mean ± SEM (n = 3) for (**B**–**E**). * *p* < 0.05 vs. Na^+^ (0), K^+^ (10) or Na^+^ (70), K^+^ (10) for (**C**) and vs. K^+^ (10), CCCP (0) or K^+^ (70), CCCP (5) for (**E**).

**Figure 4 ijms-25-01233-f004:**
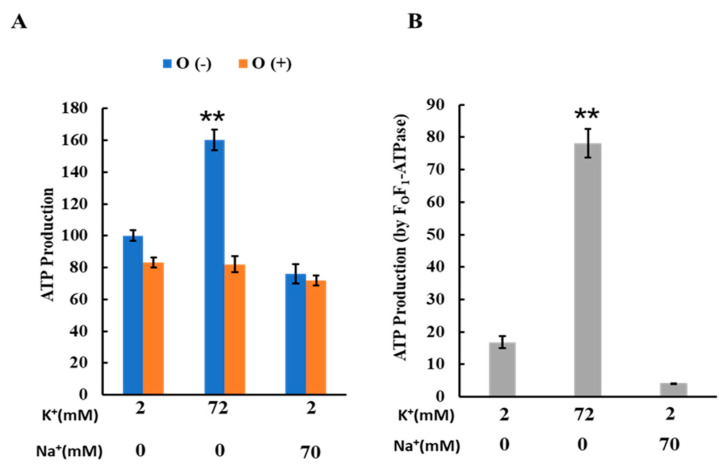
Effects of K^+^ and Na^+^ on ATP production: (**A**) ATP production by isolated mitochondria. ATP was produced in three different buffer conditions: 2 mM K^+^, 72 mM K^+^, and 2 mM K^+^ + 70 mM Na^+^. O (-) and O (+) are ATP production in the absence and presence of oligomycin, respectively. When oligomycin was added, oligomycin was present at 1 μΜ; ATP production in a buffer containing 2 mM K^+^ without oligomycin was set to 100. (**B**) ATP production by FoF_1_-ATPase in isolated mitochondria; ATP production by FoF_1_-ATPase was calculated as the difference between ATP production in the presence and absence of oligomycin. Three independent experiments were performed for each condition. Values represent the mean ± SEM (n = 3). ** *p* < 0.01 vs. buffer with 2 mM K^+^.

**Figure 5 ijms-25-01233-f005:**
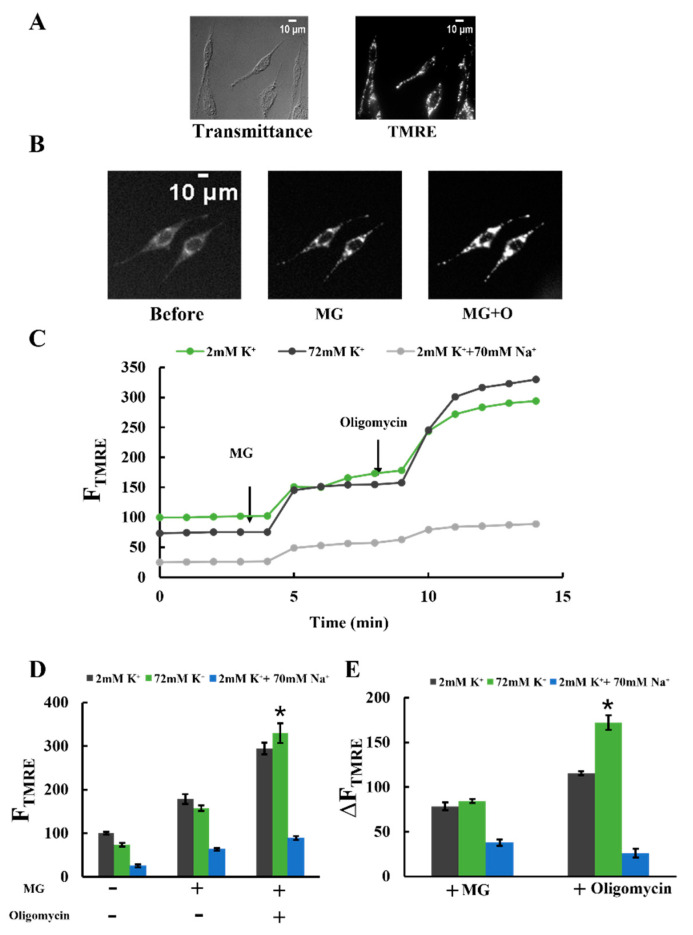
Effects of K^+^ and Na^+^ on mitochondrial membrane potential: (**A**) transmittance and TMRE fluorescence images of plasma membrane-permeabilized C6 cells. The transmittance image was obtained using differential interference optical microscopy. Bar, 10 μm. (**B**) Changes in TMRE fluorescence images in a Tris-70K buffer with 1 mM KH_2_PO_4_, 0.5 mM KH_2_ADP, and 1 mg/mL BSA. TMRE fluorescence images were obtained before and after adding 5 mM malate and 5 mM glutamate (MG) and after further adding 1 μΜ oligomycin (MG + O). Bar, 10 μm. (**C**) Time-resolved TMRE fluorescence changes of plasma membrane-permeabilized C6 cells. MG and oligomycin were added at t = 4 and 9 min, respectively. TMRE fluorescence was measured in three different buffer conditions, as explained in Figure 4A. Each buffer contained 1 mM KH_2_PO_4_ and 0.5 mM KH_2_ADP. The average value of F_TMRE_ in a 2 mM K^+^ buffer before adding malate and glutamate (MG) was set to 100. (**D**,**E**) TMRE fluorescence (**D**) and TMRE fluorescence changes (**E**) upon adding MG and oligomycin in plasma membrane-permeabilized C6 cells. A total of 3 independent experiments were performed for each condition, with 15 cells analyzed in each experiment. Values represent the mean ± SEM (n = 3). * *p* < 0.05 vs. 2 mM K^+^ for respective conditions.

**Figure 6 ijms-25-01233-f006:**
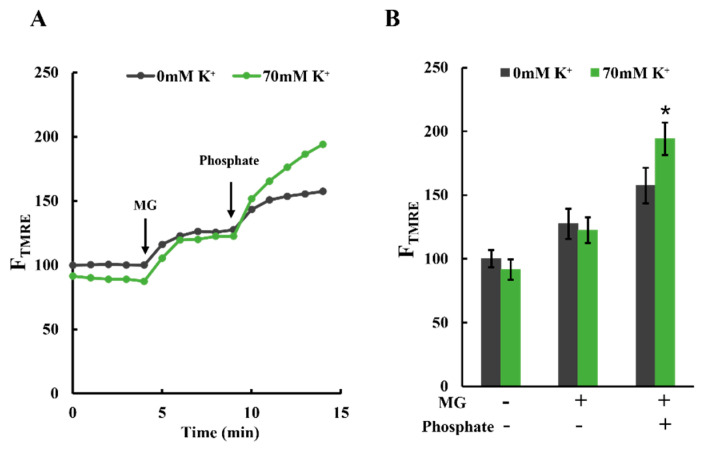
Effects of K^+^ on phosphate-induced polarization of mitochondria: (**A**) time-resolved TMRE fluorescence changes. Plasma membrane-permeabilized C6 cells were incubated in Tris-0K buffer or Tris-70K buffer. TMRE fluorescence of cells in Tris-0K buffer at t = 0 was set to 100; 5 mM malate and 5 mM glutamate (MG) were added at t = 4 min, and 1 mM phosphate (KH_2_PO_4_) was added at t = 9 min. (**B**) Statistical analysis of TMRE fluorescence in plasma membrane-permeabilized C6 cells. More than 3 independent experiments were performed for each condition, with 15 cells analyzed in each experiment. Values represent the mean ± SEM (n > 3). * *p* < 0.05 vs. 0 mM K^+^ for each condition.

**Figure 7 ijms-25-01233-f007:**
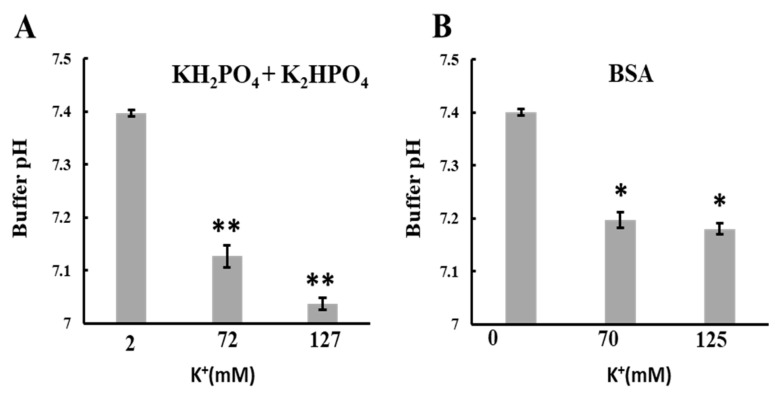
Effects of KCl on the pH of buffer or BSA solution: The pH of the solution was measured using a glass electrode at 25 °C. A KCl solution in ultrapure water was added to the buffer or BSA solution and adjusted to pH 7.4 to obtain 70 or 125 mM KCl. Ultrapure water without KCl was added to the buffer as a control. The solutions used were (**A**) KH_2_PO_4_ + K_2_HPO_4_ (1 mM PO_4_, 2 mM K^+^) and (**B**) 10 mg/mL BSA. Values represent the mean ± SEM (n > 3). ** *p* < 0.01; * *p* < 0.05 vs. 2 mM K^+^ for (**A**) or 0 mM K^+^ for (**B**).

**Figure 8 ijms-25-01233-f008:**
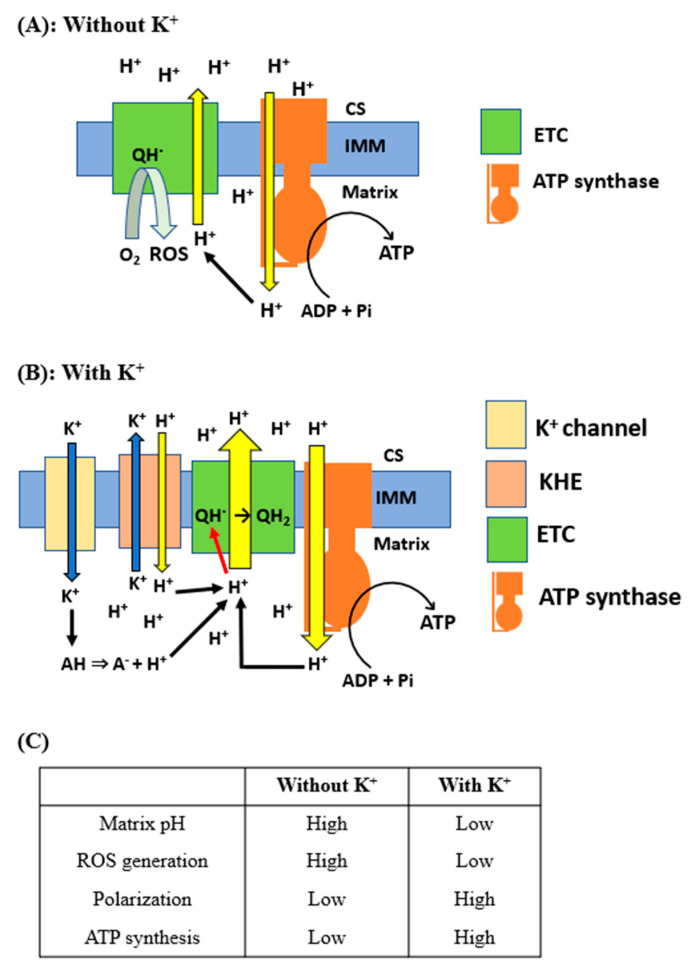
Proposed model for the effects of K^+^ on mitochondrial ATP and ROS production: The schematic illustration of the proposed model for the effects of K^+^ on mitochondrial matrix pH, membrane potential, and production of ATP and ROS is shown. (**A**) Supply and translocation of H^+^ in the absence of K^+^. CS, cristae space; IMM, inner mitochondrial membrane; QH^·^, ubisemiquinone. (**B**) Supply and translocation of H^+^ in the presence of K^+^. AH, weak acids; QH_2_, ubiquinol. (**C**) Comparison of mitochondrial function with and without K^+^.

## Data Availability

The original contributions presented in the study are included in the article and Supplementary Material; further inquiries can be directed to the corresponding authors.

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
