# Peer review of "Potassium Ions Decrease Mitochondrial Matrix pH: Implications for ATP Production and Reactive Oxygen Species Generation"

_ijms, 2024, doi:10.3390/ijms25021233_

Round 1

Reviewer 1 Report

Comments and Suggestions for Authors

The paper is s well-structured and informative, but there are a few potential areas for improvement or clarification:

1.       The abstract mentions a reduction in matrix pH with increasing extramitochondrial K+ concentrations. It would be helpful to explicitly state whether this reduction is considered beneficial, detrimental, or if the implications are not clear.

2.       While the abstract mentions that elevated K+ levels trigger mitochondrial polarization, it might be beneficial to provide a brief explanation or context for readers unfamiliar with the term. This could enhance the accessibility of the abstract.

3.       The abstract mentions the use of fluorescence imaging techniques, but it could benefit from a brief overview or mention of the specific techniques used. This would provide readers with a better understanding of the reliability and precision of the measurements.

4.       The abstract mentions a decrease in ROS generation with increasing extramitochondrial K+ concentrations. A brief explanation or speculation on the potential implications of this reduction could add depth to the understanding of the study's findings.

5.       While the abstract mentions that elevated K+ levels boost ATP synthesis via FOF1-ATPase, a brief clarification or mention of the magnitude of this boost or its physiological relevance would enhance the impact of the conclusion.

6.       It might be beneficial to include a sentence acknowledging any limitations of the study or suggesting potential avenues for future research. This could provide a more balanced perspective on the findings.

7.       The abstract could briefly discuss the broader implications of the study's findings in the context of cellular health or potential applications in medical or biological research

8.       The use of certain phrases and terminology is repeated throughout the introduction, such as "sub-hundred millimolar" and "mitochondrial matrix." While it is essential to convey key concepts, varying the language and introducing synonyms could enhance readability.

9.       The introduction discusses conflicting findings on the effects of K+ on ATP synthesis and ROS generation, but it might benefit from a concise summary or explanation of the existing literature's divergent results. This could provide readers with a better context for the current study.

10.   The introduction mentions the study's methodology involving the introduction of K+ into the mitochondrial matrix under mild conditions. Providing a brief explanation of these conditions or their relevance to the research objectives would enhance clarity.

11.   While the introduction cites general references for background information, it would be beneficial to include more specific references for certain statements, especially those related to conflicting findings in the literature. This adds credibility to the claims made in the introduction.

12.   The focus of the introduction is on ATP synthesis and ROS generation, but mitochondria have various other functions. A brief acknowledgment or justification for narrowing the focus to these specific aspects could be included.

13.   The transition from discussing the general background to introducing the study's methodology and findings could be smoother. A sentence explicitly stating the study's contribution or objective would help readers understand the specific focus of this research.

14.   While some abbreviations are defined (e.g., ROS, ATP), others like KHE and FOF1-ATPase are not explained. Providing definitions upon first use or in a separate section could aid readers unfamiliar with these terms.

15.   Add appropriate cat# for all consumables and reagents with manufacturer details

16.   Mention p value on the figure itself

17.   Figure legends are too lengthy especially figure 8. Highlight the findings and present in box and keep on the corner of figure/s

Comments on the Quality of English Language

Minor editing required

Author Response

Thank you very much for your comments. 

Please take a look at the file I've attached.

Reviewer 2 Report

Comments and Suggestions for Authors

This is a well-designed study that addresses the fairly traditional topic of mitochondrial potassium homeostasis. Indeed, I agree with the authors that there are many unclear points in this area that require attention. I had a few comments about the design of the work:

1. The authors mention that the physiological concentration of K+ is about 140 mM. The question immediately arises as to why the authors used about 70 mM K+ in most experiments. Of course, Fig. 1 and 2 show data with 125 mM K+ and in these cases there is practically no further change in the assessed parameters. However, this may only apply to ROS generation and pH changes. I recommend that the authors use more physiological concentrations of K+ also in parts 3.3 – 3.6, in order to eliminate any doubts that arise in the potential reader, and also confirm their hypothesis voiced at the end of the discussion section.

2. Part 3.3. Lines 288-289. Can the authors test the effects of other mitochondrial potassium channel inhibitors to discuss their role in the observed effects?

3. Is it possible to confirm the role of K+/H+ exchanger in the observed effects? For example, using inhibitory analysis.

4. I invite the authors to present the results presented in Part 3.7 as supplementary material.

Author Response

(The authors gave the same response as above.)

Reviewer 3 Report

Comments and Suggestions for Authors

The present work highlights the important role of mitochondrial K+ in maintaining cellular health and providing solid experimental evidence of the ability of K+ to reduce the pH within the matrix, suppress ROS generation and stimulate the energy efficiency of mitochondrial synthesis of ATP by oxidative phosphorylation.

The reviewer has no criticisms to detect in the content of the manuscript that results overall suitably structured and providing informative and adequate inputs into the field of study. The research design resulting appropriate, the methods valid and adequately described, and the results clearly presented. The quality of the figures is suitable and the reference list covers the relevant literature satisfactorily.

The reviewer evaluates this study as appropriate for publication and acceptable in the present form.

Author Response

Thank you for taking the time to peer review our paper. It was very encouraging for us to see that our arguments were understood.

Reviewer 4 Report

Comments and Suggestions for Authors

In this study, Naima and Ohta investigate the effect of K+ ions on matrix pH and superoxide production in isolated mitochondria, precisely membrane-permeabilized cells. They provide useful additions concerning the many factors that influence mitochondria and their manifold functions. In general, the authors provide rational and solid evidence by using different experimental approaches und techniques.

From the experimental site a few additions, especially concerning the ROS measurements should be made to complete the already compelling story (see major experimental points).

Overall the manuscript is excellently written and understandable. The interpretations of the data were accurate and reasonable. I have no further corrections for the text.

Author Response

Thank you for your comments.

Please see the file I have attached.

Round 2

Reviewer 2 Report

Comments and Suggestions for Authors

The authors adequately responded to most of my comments, although I think the first point is critical and not adequately addressed. Insufficient time to supply the reagent is not a scientifically based reason; I recommend that the authors refrain from such answers.

Author Response

Thank you for your comments. 

Reviewer 4 Report

Comments and Suggestions for Authors

The authors adressed all my suggestions. Thank you very much.

Author Response

Thank you very much for your review.

We appreciate that your helpful comments were very useful for revising this manuscript.